# Autonomous Robotic System to Prune Sweet Pepper Leaves Using Semantic Segmentation with Deep Learning and Articulated Manipulator

**DOI:** 10.3390/biomimetics9030161

**Published:** 2024-03-05

**Authors:** Truong Thi Huong Giang, Young-Jae Ryoo

**Affiliations:** 1Department of Information Technology, Tay Nguyen University, Buonmathuot 63161, Vietnam; tthgiang@ttn.edu.vn; 2Department of Electrical and Control Engineering, Mokpo National University, Muan-gun 58554, Republic of Korea

**Keywords:** autonomous robotic system, robotic pruning, agricultural robotics, 3D point clouds, semantic segmentation neural network, sweet pepper

## Abstract

This paper proposes an autonomous robotic system to prune sweet pepper leaves using semantic segmentation with deep learning and an articulated manipulator. This system involves three main tasks: the perception of crop parts, the detection of pruning position, and the control of the articulated manipulator. A semantic segmentation neural network is employed to recognize the different parts of the sweet pepper plant, which is then used to create 3D point clouds for detecting the pruning position and the manipulator pose. Eventually, a manipulator robot is controlled to prune the crop part. This article provides a detailed description of the three tasks involved in building the sweet pepper pruning system and how to integrate them. In the experiments, we used a robot arm to manipulate the pruning leaf actions within a certain height range and a depth camera to obtain 3D point clouds. The control program was developed in different modules using various programming languages running on the ROS (Robot Operating System).

## 1. Introduction

Sweet pepper is a valuable vegetable crop that is grown worldwide. It contains high amounts of vitamins A, B, and C, as well as several minerals [1]. Nowadays, sweet peppers are cultivated in greenhouses to provide fruits throughout the year, even in winter. However, fruit quality is greatly influenced by environmental factors and pruning methods [1,2,3,4]. Sweet pepper plants produce numerous leaves throughout the year, from the roots to the tips, which farmers usually remove manually, a repetitive and time-consuming task. Therefore, we decided to investigate an automated system for this purpose in our research.

Autonomous robots to apply agriculture is an emerging technology. It is considered to be the key to increasing productivity while reducing the need for human labor. Robots are capable of performing repetitive tasks and can operate at any time, whether it is day or night. Autonomous robots are utilized for harvesting apples, sweet peppers, cucumbers, strawberries, and tomatoes [5,6,7,8,9,10,11,12,13]. They are also used for pruning tomato plants, apple trees, and grape vines [14,15,16,17,18]. Although they may serve different purposes and have varying hardware structures, autonomous robotic systems must consist of three fundamental modules: vision perception, action point (for cutting or picking) in 3D space, and manipulation. In the case of harvesting sweet peppers, neural networks combined with 3D information are utilized to detect peduncles [6,7]. For pruning fruit trees, the aim is to detect tree skeletons and pruning positions on 2D images using deep learning techniques and then obtain 3D positions based on depth images. It is not necessary to reconstruct the entire tree in 3D since it consumes more time and resources. To recognize branches, mask R-CNN [19] was employed with multiple backbones models [14,16,18]. This pruning method is quite successful with fruit trees, but applying the same technique to greenhouse plants such as sweet peppers or tomatoes is challenging. The pruning of fruit trees is carried out in the winter when there are no leaves on the trees, making it easier to detect the skeletons without any occlusions or hidden parts. In addition, all the branches can be cut during pruning. Otherwise, greenhouse plants have many leaves, which makes it difficult to detect the parts that need pruning. Moreover, only leaves below a specific height or the first fruit should be cut. Hence, a 3D reconstruction based on multiple RGB-D images is a helpful technique to obtain the plant’s structure and reduce information loss. This result helps in implementing different pruning strategies easily. Once the system determines the pruning positions, it needs to maneuver a pruning tool to the target. This operation requires high precision and collision avoidance. To meet these requirements, the robot arms with 5 or 6 degrees of freedom are employed [6,7,10,12,14,18].

In this paper, we propose an autonomous robotic system to prune sweet pepper leaves that involves three tasks: the perception of crop parts, determination of pruning area, and manipulation of the robot arm. We utilized a semantic segmentation neural network to recognize the parts of sweet pepper crops on 2D images and generate semantic segmented images. We then created 3D semantic point clouds based on RGB-D images and semantic segmented images to detect the pruning position and generate the pruning direction. Eventually, the articulated manipulator can approach the pruning position and the end-effector performs precision pruning. To demonstrate our method, we chose simple pruning rules, which were to cut the lowest leaf of a stem. We used an Intel RealSense to obtain RGB-D images and a Universal Robot UR3 with an end-effector to perform the pruning action. The system is complex and comprises several modules. Our previous research has focused on semantic segmentation neural networks to recognize plant parts and how to build 3D semantic point clouds based on sequences of RGB-D images [20,21]. In this paper, we present a method to find the pruning position and direction from the 3D semantic point clouds, as well as the controlling robot arm program and an end-to-end autonomous pruning system.

## 2. Related Works

### 2.1. Recognition of Sweet Pepper Crop Parts

As stated earlier, perception plays a crucial role in providing the system with information about the location and nature of crop parts on RGB images. This process involves two tasks, namely classification and segmentation. In the early 2010s, several semantic segmentation algorithms were implemented, but they failed to meet the required standards [22,23,24,25]. With the development of deep learning, there has been significant progress in semantic segmentation. Deep Convolutional Neural Networks (DCNNs) are used to extract features, while Fully Connected Neural Networks are used to classify objects. These semantic segmentation neural networks can classify objects at the pixel level and are composed of two parts: the encoder and decoder. The encoder typically includes a backbone neural network consisting of a long line of DCNNs used to extract image features at various resolutions. To improve the efficiency of neural networks and reduce computational resources, new types of convolutions, such as dilated convolution (Atrous convolution) [26] and depthwise separate convolution [27], have emerged. Some of the popular convolutional neural network backbones used in computer vision tasks include VGG-16 [28], ResNet [29], Xception [30], MobileNetV2 [31], and MobileNetV3 [32]. These networks have shown improved performance by incorporating new convolutional techniques. The DeepLab model has also demonstrated the effectiveness of Atrous Spatial Pyramid Pooling, which utilizes multiple parallel Atrous convolutional layers with varying scales to enhance the model’s performance [26]. Building on these successes, we proposed a real-time semantic segmentation neural network that recognizes different parts of greenhouse plants [20]. This neural network comprises bottleneck blocks, which are introduced in MobileNetV2, a pyramid pooling block to obtain features at multiple scales. Depth images were also explored to improve the network performance.

We prepared a dataset consisting of approximately 1000 images of sweet peppers to train the neural network. The images were captured using an Intel RealSense L515 camera from various greenhouses, and each image was expected to have a stem and other relevant parts present. To annotate the images, we utilized the Semantic Segmentation Editor software v1.5. An example from this dataset is shown in Figure 1 to provide a better understanding. The stems, leaves, petioles, and fruits are represented in purple, green, pink, and red colors in the right image of Figure 1. Black color represents other objects. This type of image is called a semantic image.

### 2.2. Three-Dimensional Point Cloud Creation of Sweet Pepper Crop

Three-dimensional reconstruction is the process of creating a 3D representation of objects from a set of 2D RGB or RGB-D images. The output of this process is 3D point clouds. One of the important techniques for 3D recovery is Structure from Motion (SfM) recovery [33]. In SfM, feature points in a pair of RGB images are first extracted to match the objects, and then camera poses are estimated to create 3D sparse point clouds. Some popular algorithms for extracting feature points include SIFT [34], SURF [35], FAST [36], and ORB [37]. The Bundle Adjustment (BA) algorithm is used to improve the accuracy of the camera poses and 3D point positions simultaneously [38]. However, creating a dense point cloud still posed difficulties until the RGB-D camera became popular. The combination of SfM and RGB-D images has achieved promising results in 3D reconstruction, but it cannot run in real time. Visual SLAM is a camera-based sensor system that performs simultaneous localization and mapping in real-time, requiring fewer resources and being suitable for robotics [39]. The ORB-SLAM family [40,41,42], which includes ORB-SLAM3, is a popular open-source visual SLAM system that supports various types of cameras, such as monocular, stereo, and RGB-D cameras with pin-hole or fisheye lens models. It uses the ORB and BA algorithms to extract feature points and optimize camera poses in local or global maps. Due to its advantages, we proposed a method to create 3D point clouds by using ORB-SLAM3 and then optimizing the result with the help of the Iterative Closest Point (ICP) method [21]. The process is described in Figure 2.

In this method, one camera is used to obtain RGB-D images. The camera is moved around the plant to obtain RGB-D images. These images are firstly the input of ORB-SLAM3 to find the camera pose. At the same time, these images are passed through a semantic segmentation neural network to recognize plant parts and create semantic images. We did not create a dense map of the sweet pepper and its surrounding objects. Once the camera pose and semantic image are obtained, a 3D semantic point cloud is generated that includes only parts of the tree. The camera pose, which is retrieved from ORB-SLAM3 in this way, cannot be refined by ORB-SLAM3 when it performs a loop. Furthermore, the drift problem is cumulative from one frame to another. The ICP method is used to register each point cloud to reduce this error. After conducting many experiments, we found that moving the camera in a straight trajectory is recommended for the best results. It is also suitable for greenhouse working environments because the robot always runs on rails between rows of sweet peppers. 

We used a semantic segmentation neural network to select only the sweet pepper crop parts. This technique helps reduce the number of points in the point clouds, ultimately saving time. The final output is 3D semantic point clouds of the sweet pepper obtained from multiple RGB-D images.

## 3. Proposed Autonomous Robotic System

### 3.1. Pruning Position Detection

In our previous research, we introduced a technique for identifying pruning positions by detecting the intersection of a petiole and a stem on 2D semantic images. We then located the pruning regions by generating 3D semantic point clouds. The center point of a pruning region is considered a pruning position [21]. This method has some drawbacks. Firstly, the pruning point is detected in some different 2D images with different viewpoints. Therefore, when projecting them in 3D space, they are not always close to each other for making a perfect pruning group of points. Second, the pruning point is detected by the intersection of a petiole and its stem when they are enlarged on 2D images. The distance from the pruning point to the stem is not stable. It can be very close or far from the stem. Finally, the process of finding pruning points on the 2D image must be performed iteratively until the 3D point cloud generation process is finished. So, this method has high time consumption and the risk of damaging stems in cutting leaves due to the proximity of the pruning positions to the stems. In this study, we propose a new method for detecting pruning positions, as shown in Figure 3. The important difference in this new method is that we find the cutting point after the process of creating the 3D point cloud is finished, and the distance from the cutting point to the corresponding stem is precisely calculated.

To start, we use a previously established algorithm [21] to extract individual petiole point clouds from the 3D semantic point clouds. Next, the petioles are sorted in ascending order of height by determining their center points. To achieve our research objective, we select the lowest petiole to prune. With each petiole center point denoted P_c_ (x_c_, y_c_, z_c_), we detect a group of stem points having Oz values within z_c_ ± 0.01 m. After removing noise points, the center point of this stem group is used to detect the distance between the stem and the candidate pruning points. The selected petiole is then divided into five equal segments based on the O_x_ value within the 3D coordinate system of the robot arm, and we represent each segment by its respective center point. The final pruning position is determined by selecting the point closest to the corresponding stem, provided that it is at a distance greater than a specific threshold, denoted as “t”. After conducting experiments, we determined that a threshold of 0.02 meters is the optimal distance as it ensures that the end-effector remains clear of any potential stem collisions. In addition, this distance should be less than 0.1 m to be sure that the petiole being pruned belongs to the main stem. By removing the detecting pruning point process on 2D images, the 3D semantic point cloud-creating process takes 0.2–0.3 s for one 640 × 480 RGB-D image, while it took 0.725 s with the previous method.

### 3.2. Pruning Direction Estimation

After detecting the pruning positions, the following step is to establish a rotation matrix that aligns the robot arm with the desired pruning direction. For obtaining the ideal cut, as shown in images Figure 4b,c, it is crucial to ensure that the end-effector is perpendicular to the petiole. Our approach involves a four-step process for determining the rotation matrix. Firstly, we identify the petiole vector, $\vec{v1}$, which is defined by the pruning position and one of the five petiole positions. It should be the closest point to the pruning point. Secondly, using the positions of the camera and pruning position, the camera $\vec{v2}$ is determined. In the third step, the perpendicular vectors $\vec{v3}$ and $\vec{ve}$ are computed as detailed in Equation (1). The combination of $\vec{v3}$, $\vec{v1}$, and $\vec{ve}$ corresponds to the Ox, Oy, and Oz coordinate axes of the end-effector, collectively forming the precise end-effector pose. Finally, we normalize all vectors, after which the rotation matrix of the end-effector can be calculated, as shown in Equation (2).
(1)$$\vec{v3}=\vec{v1}*\;\vec{v2}\;\vec{ve}=-\vec{v3}*\;\vec{v1}$$
(2)$$\left[\begin{array}{ccc} {v3}_{x} & {v1}_{x} & {ve}_{x}\\ {v3}_{y} & {v1}_{y} & {ve}_{y}\\ {v3}_{z} & {v1}_{z} & {ve}_{z} \end{array}\right]$$

### 3.3. Articulated Manipulator

The manipulator comprises a 6-degree-of-freedom robot arm and a gripper, which is known as an end-effector. To control the robot arm, the open-source software framework MoveIt [43] is used. It provides a comprehensive set of tools and libraries that enable robots to generate motion-plan paths, visualize, avoid collision, and execute movements. Additionally, MoveIt has a 3D perception function that enables the robot to perceive its environment. This is important for collision detection and obstacle avoidance during motion planning. To prevent collisions with stems and fruit, their point clouds are configured as collision objects. Once the pruning pose is received, the RRT-Connect algorithm [44] is applied to find collision-free paths from the start position to the pruning position. The pruning process is described in Figure 5 as the module manipulation. Firstly, the robot is initialized in the ready position when the robot arm is up and 0.5 m away from the plant, as in Figure 5. Then, the pruning direction module calls services and passes the pruning direction to the manipulation module. At this time, MoveIt detects the collision objects and generates the moving path for the robot arm. The end-effector is maneuvered to the first position, and then the cutter is opened and moved to the pruning position in the same direction. Finally, the pruning action is performed, and the robot moves back to the first position.

### 3.4. Autonomous Robotic System for Pruning

The system is composed of three main modules, each written in a different programming language and separated from the other. These modules include the image perception module, which uses a semantic segmentation neural network to detect crop parts; the pruning direction module, which reconstructs the crop in 3D space and detects the pruning direction; and the robot controlling module, which moves the robot arm to the pruning position and performs the pruning action. ROS is used to employ ROS topics and ROS services to connect these modules [45]. Figure 6 illustrates the structure of the entire system. ROS offers a flexible working environment, facilitating the connection and modification of system components. Figure 5 shows the activity diagram of the whole system, from obtaining RGB-D images to pruning action. 

## 4. Experiments and Results

### 4.1. Training the Semantic Segmentation Neural Network

The training program was written in Python 3.9 and uses PyTorch 1.8 and Torch-Vision 1.10 library. The model was trained and tested on a machine with GPU Nvidia Ge-force RTX 3090 with CUDA Version 11.2. The model has the best IOU (intersection over union) of 0.69 and fps (frame per second) of 138.2 after 25 epochs. Figure 7 shows the loss and IOU in the training and evaluating process of the mode, and Figure 8 shows an example of plant part prediction of the semantic segmentation neural network.

### 4.2. Running the Robotic System

Our hardware setup is shown in Figure 9. The robotic system consists of a 6-degree-of-freedom Universal Robot UR3, an end-effector OnRobot RG2, which represents a cutting end-effector, and an Intel RealSense L515. The image perception module was written in Python, and other modules were written in C++. These modules are written as ROS Node of ROS Noetic under Ubuntu 20.04.6. They can interact with each other via ROS service and ROS topics.

These experiments focus on the performance of the whole system, including the percentage of success, failure, and failure reasons. The experiments were conducted in the lab. The robotic system had to prune the lowest leaf. Table 1 shows the results of the experiments. We tried 30 times with different plants and positions. Overall, 57% of attempts were successful, and 43% of attempts failed, in which 20% of failures was due to out-of-reach or inability to find the collision-free paths, 13.33% could not detect the pruning position, and 6.67% detected the incorrect pruning positions, and 3% of robot arms encountered environmental obstacles due to not being detected in the semantic point cloud.

This result shows that the whole system can archive the pruning task but not in high performance. There are three main failure reasons. Firstly, the system is unable to identify the lowest pruning position as per the experimental requirements. This is due to the objects being hidden from view. In certain cases, the lowest leaf is obscured by other leaves, while in others, the petiole is too small for the camera to capture depth information. Secondly, 20% of the experiments failed because the pruning positions were out of reach or there were obstacles in the robot’s path, preventing MoveIt from generating motion plan paths. The UR3’s standard reach is approximately 500 mm, and the end-effector is about 230 mm, which limits the robotic system’s maximum reach. Additionally, the camera used in the experiments was an obstacle object and was positioned close to the robot arm. Finally, a small number of errors occurred due to inaccuracies in semantic segmentation prediction, resulting in errors in pruning position detection.

The performance of the system can be improved by letting the robot arm and camera on a mobile robot. Our 3D reconstruction modules can create 3D point clouds from many RGB-D images, but in these experiments, the camera cannot move. Therefore, if the robot arm and camera are placed on a mobile robot to move through these crops to take images and create 3D point clouds before detecting the pruning position, it can reduce the problem of occlusion and hiding. Moreover, if MoveIt cannot find the free-collision-free path, the whole system can move to a new nearby position and try to find a new one. 

The process of generating robot arm moving paths also affects the system performance. If the moving path is too long, it will take time. It should have some constraints of working safe area because one camera cannot detect all obstacles around the robot’s arm. Furthermore, the camera itself is an obstacle object too. Therefore, the position of the camera should be carefully considered. It should minimize the impact of moving path generation but also obtain the most information about the plant. 

The experiment results show that being “out of reach” is the most common error. To increase the reach of the robot arm, we can employ other longer robot arms such as UR5 or UR10. Developing a specialized robot to perform pruning operations is also promising research. 

## 5. Conclusions

We proposed an autonomous robotic system to prune sweet pepper leaves, which consists of three modules: perception, pruning position detection, and manipulation. The perception module employed the semantic segmentation neural network to recognize sweet pepper plant parts. The pruning position detection module uses 3D reconstruction and the proposed pruning pose detection methods that reduce the 3D semantic point cloud generation time from 0.7 s of the previous method to 0.2–0.3 s. We used Moveit to generate the moving path and control the robot arm in the manipulation module.

The robotic system demonstrates a complete process that takes RGB-D images as input and performs pruning actions as output. The experiments show that the proposed system performs well and has potential for future improvements. This paper also presents an overview of a smart agricultural robotic system designed for tending to greenhouse crops. The structure of this system can be applied to other crops and purposes as well.

## Figures and Tables

**Figure 1 biomimetics-09-00161-f001:**
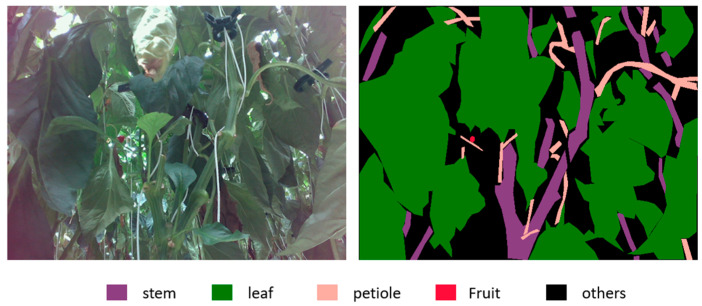
Example of annotated images.

**Figure 2 biomimetics-09-00161-f002:**
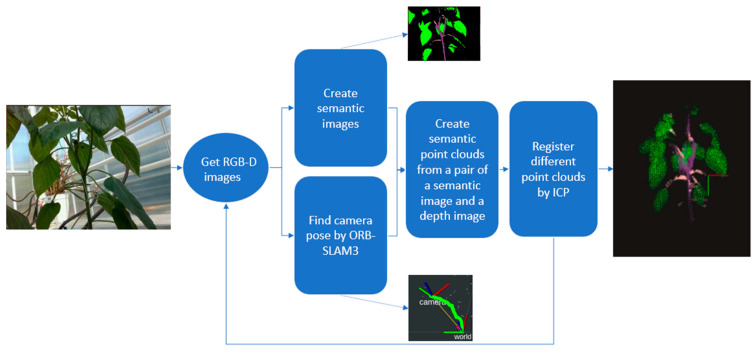
Three-dimensional semantic point cloud creation process.

**Figure 3 biomimetics-09-00161-f003:**
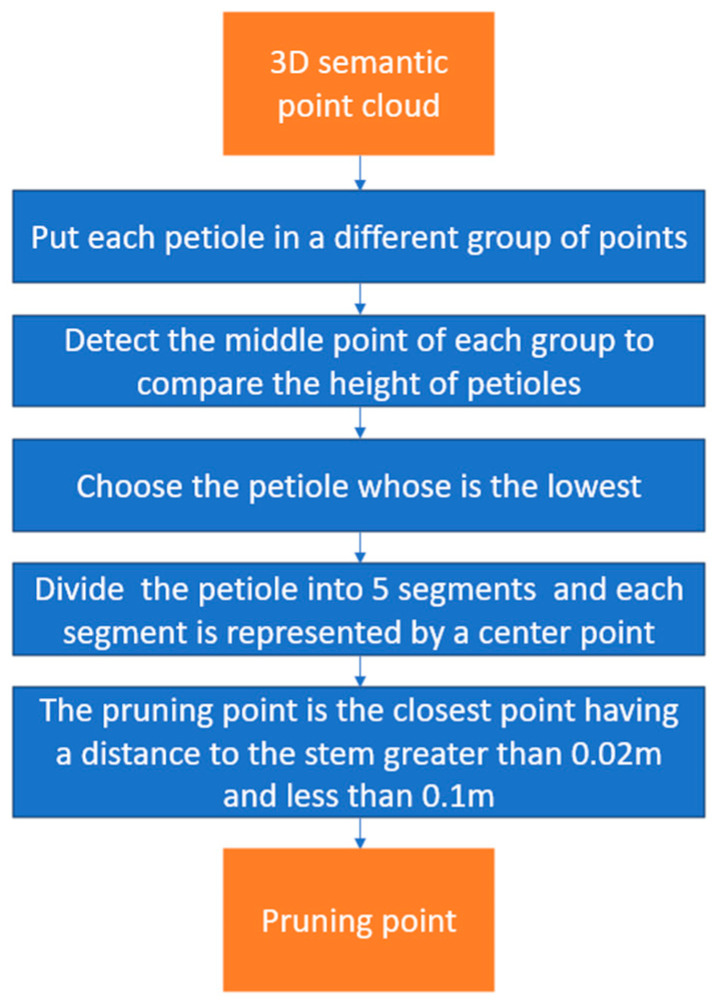
Pruning position detection process.

**Figure 4 biomimetics-09-00161-f004:**
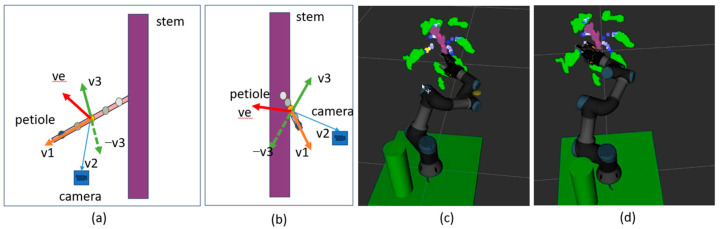
Pruning direction estimation. Images (**a**,**b**) describe how to detect an end-effector pose from the 3D semantic point clouds. Image (**a**) is the front view, and image (**b**) is the left-side view. Images (**c**,**d**) describe how an end-effector prunes a petiole in a simulator environment. The five blue points from light blue to deep blue are five petiole points. The yellow point is the pruning position.

**Figure 5 biomimetics-09-00161-f005:**
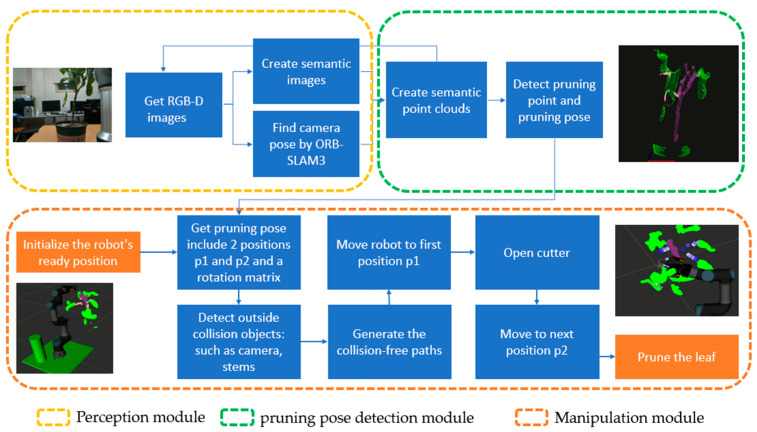
Activity diagram of the whole system.

**Figure 6 biomimetics-09-00161-f006:**
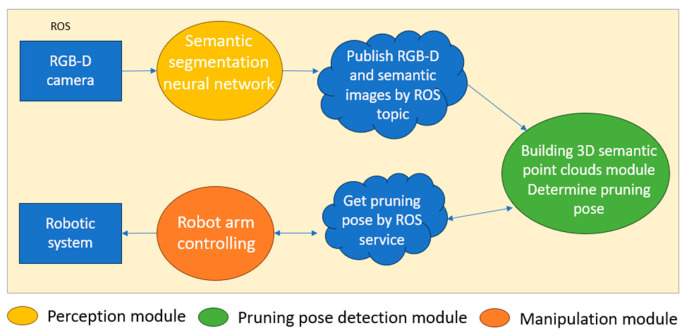
Structure of autonomous robotic system for pruning.

**Figure 7 biomimetics-09-00161-f007:**
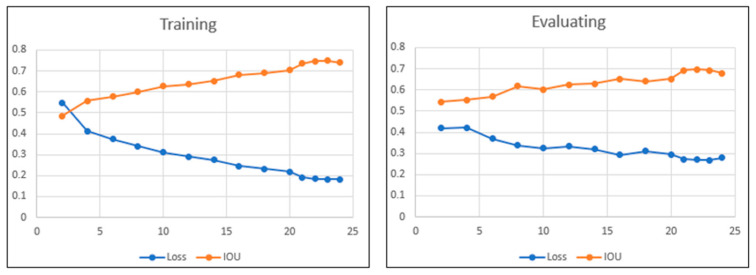
The loss and IOU value of training and evaluating process.

**Figure 8 biomimetics-09-00161-f008:**
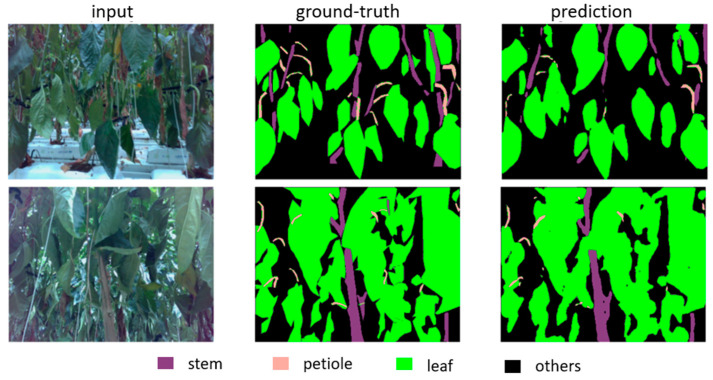
The result of the semantic segmentation neural network model.

**Figure 9 biomimetics-09-00161-f009:**
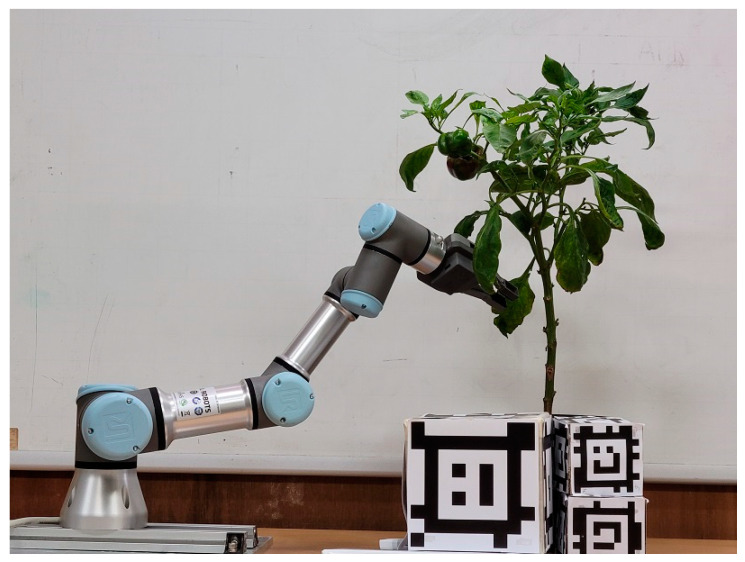
Experimental setup.

**Table 1 biomimetics-09-00161-t001:** Experiment results.

Percentage	Result	Failure Reasons
57.00%	success	
13.33%	Failure	Cannot detect the pruning positions and pruning directions
20.00%		Cannot find motion plan paths or out-of-reach
6.67%		Detect incorrect pruning positions
3.0%		Hit obstacles

## Data Availability

The data presented in this study are available on request from the corresponding author (the data will not be available until the project is finished).

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
