# Peer review of "Autonomous Robotic System to Prune Sweet Pepper Leaves Using Semantic Segmentation with Deep Learning and Articulated Manipulator"

_biomimetics, 2024, doi:10.3390/biomimetics9030161_

Round 1

Reviewer 1 Report

Comments and Suggestions for Authors

see pdf

Comments on the Quality of English Language

English language is fine. Only minor issues detected.

Author Response

Thanks for the reviewer's comments.

We do our best to revise the manuscript.

Reviewer 2 Report

Comments and Suggestions for Authors

This manuscript is relevant and interesting, it is devoted to a robotic system for pruning sweet pepper leaves based on image processing using a neural network.

The manuscript is recommended for printing in this journal, but to improve the manuscript, I recommend paying attention to some points.

1. The manuscript indicates that the dataset was collected in various greenhouses, please indicate the names of the varieties of pepper from which the dataset was removed.

2. Although you have indicated links to other manuscripts related to this study concerning the neural network, in this manuscript I recommend adding graphs and calculation of well-known metrics for the quality of the neural network (Precision, Recall, F1-score), analysis of the loss function depending on the learning epoch.

3. It is proposed to expand the conclusion and compare the results obtained with other similar works,

4. It is also recommended to present the results of the full-scale experiment in more detail.

Author Response

(The authors gave the same response as above.)
